# Genetic Control of GCF Exudation: Innate Immunity Genes and Periodontitis Susceptibility

**DOI:** 10.3390/ijms241814249

**Published:** 2023-09-18

**Authors:** Zsolt M. Lohinai, Kasidid Ruksakiet, Anna Földes, Elek Dinya, Martin Levine

**Affiliations:** 1Department of Restorative Dentistry and Endodontics, Semmelweis University, H-1088 Budapest, Hungary; lohinai.zsolt@dent.semmelweis-univ.hu; 2Department of Oral Biology, Semmelweis University, H-1089 Budapest, Hungary; ksd13rsk@gmail.com (K.R.); foldes.anna@dent.semmelweis-univ.hu (A.F.); 3Department of Restorative Dentistry, Faculty of Dentistry, Naresuan University, Phitsanulok 65000, Thailand; 4Digital Health Department, Semmelweis University, H-1094 Budapest, Hungary; dinya.elek@public.semmelweis-univ.hu; 5Department of Periodontology, University of Oklahoma Health Sciences Center, Oklahoma City, OK 73104, USA

**Keywords:** experimental gingivitis, lysine, innate immunity, genes, periodontitis, periopathogens, systemic disease

## Abstract

Chronic periodontitis is a bacterial infection associated with dentally adherent biofilm (plaque) accumulation and age-related comorbidities. The disease begins as an inflammatory exudate from gingival margins, gingival crevicular fluid (GCF) in response to biofilm lysine. After a week of experimental gingivitis (no oral hygiene), biofilm lysine concentration was linearly related to biofilm accumulation (plaque index) but to GCF as an arch-shaped double curve which separated 9 strong from 6 weak GCF responders (hosts). Host DNA was examined for single nucleotide polymorphisms (SNPs) of alleles reported in 7 periodontitis-associated genes. Across all 15 hosts, an adenine SNP (A) at IL1B-511 (rs16944), was significant for strong GCF (Fisher’s exact test, *p* < 0.05), and a thymidine SNP (T) at IL1B+3954 (rs1143634) for weak GCF provided 2 hosts possessing IL6-1363(T), rs2069827, were included. The phenotype of IL1B+3954(T) was converted from weak to strong in one host, and of the non-T allele from strong to weak in the other (specific epistasis, Fisher’s exact test, *p* < 0.01). Together with homozygous alternate or reference SNPs at IL10-1082 or CD14-260 in 4 hosts, all hosts were identified as strong or weak GCF responders. The GCF response is therefore a strong or weak genetic trait that indicates strong or weak innate immunity in EG and controllable or uncontrollable periodontal disease, dental implant survival and late-life comorbidities.

## 1. Introduction

### 1.1. Periodontal Disease

Chronic adult periodontitis is a bacterially induced inflammatory disease that slowly destroys the connective tissue and bone-supporting teeth in the oral cavity. It is preceded by gingival inflammation (gingivitis) which first appears in teenagers and young adults whose oral hygiene is inadequate. Table 1 shows that periodontitis affects 47% of the US population between age 30 and 65 [1], often accompanied by comorbidities such as diabetes, cardiovascular disease (CVD) [2], arthritis [3], dementia and cognitive impairment [4,5,6]. The reason why so many comorbidities are associated is not clear. A current hypothesis is infection of the gingival crevice with certain gram-negative bacteria, especially *Porphorymonas gingivalis* which can spread throughout the body [6].

A study of the severity of periodontitis in identical and non-identical adult twins indicates almost 50% heritability, even after adjustment for smoking and behavioral variables [7]. Correspondingly, certain variants (alleles) of genes that encode a genotype consisting of *Interleukins* 1A and 1B (*IL1A* and *IL1B*) were found to detect individuals (hosts) with tooth loss due to severe periodontitis after adjusting for smoking, diabetes and professional dental care [8]. How that genotype is linked to a host’s innate immune response that determines resistance to infections is poorly understood.

A better way of looking at genotypes associated with periodontitis could be to determine if they have a role in experimental gingivitis (EG) [9]. This procedure requires gingivally healthy volunteers (hosts) who agree to restrict all oral hygiene procedures for up to 3 weeks following a period of intensive oral hygiene. Gingivitis is measured as the gingival index (GI) or the exudation of gingival crevicular fluid (GCF), a measure of subclinical inflammation [10]. GCF increases within a week of ceasing oral hygiene, but GI requires 2–3 weeks to appear. We have reported that GCF responds to the lysine concentration extracted from dentally adherent biofilms after EG for a week [11] and subsequently that similar biofilm lysine concentrations gave strong or weak GCF responses in different hosts, https://iadr.abstractarchives.com/abstract/18iags-2958594/high-or-low-gingival-crevicular-fluid-response-after-experimental-gingivitis (accessed on 1 May 2023). 

### 1.2. Evidence for Strong and Weak Inflammatory Responses to Oral Bacteria in EG

Recent studies of EG in Danish and US cohorts identified fast and slow rates of gingivitis development associated with fast and slow inflammatory mediator development [12,13,14,15]. These reports indicate the presence of strong and weak innate immune host responses to dentally adherent microbiomes that result in inflammasome activation of the *IL1* gene family [16,17]. Strong innate immunity is detected as a rapid, strong gingivitis response that releases cytokines to activate thymus-passaged lymphocytes (T-cells) and antibody production to remove foreign material, repair damaged cells and control disease (acquired immunity). Conversely, weak innate immunity gives a slow, weak inflammatory response that results in a lack of acquired immunity.

In health, mastication intermittently damages the dentally attached (DAT) cells of junctional epithelium (JE, Figure 1 and Figure 2A), causing traces of GCF to exude from gingival crevices where it supports epithelial attachment turnover [18,19]. Dentally adherent salivary bacteria become exposed to the GCF, but variations in chemokine expression determine greater or lesser concentrations of neutrophils, and a healthy microbiome in the gingival crevice in different individuals [20]. Bacteria remaining in the crevice may cause the DAT cell attachment to migrate apically and produce a deepened crevice, a precursor of mild periodontitis (Figure 2B).

An EG study from China detailed how a strong or weak gingivitis response after 3 weeks was related to differences in microbial colonization [21]. Using ribosomal RNA sequencing, the authors identified 27 microbial genera whose abundance in biofilm significantly correlated with gingivitis severity in 50 hosts. Yet the relative abundance of genera differed in strong compared with weak gingivitis responders. Each of these genera were subsequently identified in samples of naturally occurring biofilms obtained prior to the intensive oral hygiene preceding EG. The same result was obtained from pre- and post-EG samples from a second cohort of 41 hosts with 95% accuracy [22]. The authors concluded that the different microbial structures of natural gingivitis prior to cleaning predicted strong or weak gingivitis development during EG.

**Figure 2 ijms-24-14249-f002:**
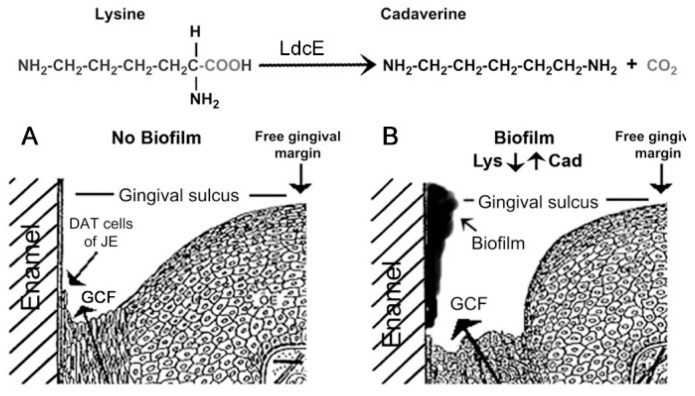
Dentally attached cells (DAT) cells and gingival crevicular fluid (GCF). Top shows the reaction catalyzed by biofilm lysine decarboxylase (LdcE). (**A**) Healthy crevice. (**B**) Infected crevice (incipient pocket). Copyright sources for Figure 2 are provided in references [11,22]: Reprinted and adapted with permission for this study in 2022 by S. Karger AG, Switzerland (Karger.com).

### 1.3. Biofilm Lysine Concentrations Determine GCF Responses after a Week of EG

As bacteria increase at the gingival margin region after EG as described in the Appendix A, *Eikenella corrodens* appear within a day or two [23]. Its outer membrane possesses lysine decarboxylase (LdcE), an enzyme that converts nutritionally essential lysine [24] to non-nutritional cadaverine (Figure 2B) [19,25]. Prior to cleaning, the mean concentration of lysine in GCF is 0.18 µmol/mL, whereas that of saliva is less than 0.01 µmol/mL [26,27], indicating that the source of biofilm lysine is GCF [19]. After a week of EG, the lysine concentration of biofilm varies from about 0.02 µmol/g biofilm to 0.22 µmol/g, but below 0.11 µmol/g biofilm in 13 of the 16 hosts Appendix A [11].

The plaque index (PI) measure of biofilm accumulation increases collinearly with lysine and GCF exudation but above 0.11 mM (0.11 µmol/g biofilm), the PI and GCF measurements diverge. PI and lysine keep increasing (Appendix A), but GCF decreases (Appendix A). The demarcation at 0.11 µmol lysine/g biofilm corresponds to the minimal concentration of lysine in blood plasma compatible with health [28]. Below that amount, DAT cells are lysine starved, causing the JE to become permeable to biofilm products. Above that critical lysine content, the basal cell layer of the JE becomes stronger and prevents bacterial product penetration. At 0.22 µmol lysine/g biofilm, Appendix A shows that GCF is almost back to baseline levels despite the increased accumulation of biofilm, whereas Appendix A shows that lysine is a significant substrate for biofilm accumulation.

### 1.4. Rationale and Aim

Although the arched curve in Appendix A indicates the variation of GCF exudation with respect to biofilm lysine content, expanding the *y*-axis (Figure 3) clearly indicates two statistically different, non-interactive GCF responses to the same lysine concentration The double-arch centered around 0.11 µg/g biofilm therefore indicates two sets of individuals, strong and weak GCF responders to similar biofilm lysine concentrations. The rationale for this study was to investigate whether differences in SNP variants of innate immunity genes previously associated with periodontitis separate strong from weak GCF responses.

Histories of infections produce distinctive frequencies of SNP alleles within genes from different populations or ethnic groups [29]. The alternate thymidine (T) residue at *IL1B*+3954 (numbered downstream from the first nucleotide encoding its IL1β protein) is associated with many inflammatory diseases, including periodontitis where periodontopathogens [30,31] enhance IL-1β expression [32]. The association with periodontitis was reported and confirmed for *IL1B*+3954(T) in a study using 117 periodontitis patients and 175 controls. The need for so many *IL1B*+3954(T) individuals and controls suggests the presence of unknown confounding SNP variants from the same or different genes [32].

Independently of *IL1B*+3954(T), the *IL1A*-889(A), encoded protein IL1α, is associated with several chronic inflammatory diseases [33,34]. The combination of SNPs in these two genes forms the tooth loss genotype, indicated with a plus sign that connects the two allelic variants, *IL1A*-889(A)+*IL1B*+3954(T). Of 5117 US patients (hosts), who were 34–55 years of age and mostly US Caucasians, 534 patients exhibited tooth loss, severe periodontitis [8]. Of the patients in this subset, 67.6% were IL-1 genotype-positive and also smokers who had either diabetes, their teeth cleaned twice or less yearly, or both. Unfortunately, we were unable to find a report of the IL-1 genotype frequency in either all 5117 patients or the 534 host subgroup. The tooth loss genotype could therefore predict severe periodontitis associated with smoking in middle aged adults, but its ability to predict moderate and severe periodontitis in the absence of smoking seems unlikely.

Our aim was to determine whether strong and weak GCF exudation responses to similar biofilm lysine concentrations were linked to SNPs of genes associated with periodontitis. If so, they could explain whether the difference in GCF exudation predicts structurally different microbiomes after EG resembling those in natural gingivitis pre-EG [21,22], and why periodontitis associates with comorbidities such as dementia.

## 2. Results

### 2.1. Host Gene Identification, Allele Distribution, and Relationship to Strength of GCF Exudation

Table 2 lists the 7 genes and 8 alleles tested in this study. A second allele of gene *IL10* (*IL10*-597) was tested but not included because it replicated *IL10*-1082. The Reference SNP (rs) Report and the aggregate allele frequency for Europeans (ALFA Project) is provided by the US National Center for Biotechnology Information (NBCI), https://www.ncbi.nlm.nih.gov/snp/rs1800587 (accessed on 14 July 2023) or https://www.ncbi.nlm.nih.gov/snp/rs1143634 (accessed on 14 July 2023), etc. for each rs number in Table 2. The reference SNP is shown first and the alternate SNP second. 

Of the 7 genes chosen because of their influence on periodontitis (Table 2), either the reference or alternative SNP can associate with greater expression and systemic inflammation. For example, hosts possessing the alternate SNP of *IL1A*-889 and *IL1B*+3954 are each hetero- or homozygous for increased expression of their respectively encoded proteins [32,33,34,35], whereas it is the homozygous reference SNP (GG) allele of *IL1B*-511 [36] which possesses that activity along with those of *IL10*-1082(TT) [37], and *CD14*-260(AA) [38]. In all three of these genes, it is the alternate allele (hetero- or homozygous) that exhibits normal expression of encoded proteins. In the remaining 3 genes in Table 2, the alternate SNP alleles of *IL6*-1363 [39], *COX2*+8473 [40] and *MMP8*-799 [41] associate with greater encoded protein expression, and more systemic inflammation as described for *IL1A*-889 and *IL1B*+3954. 

In Figure 4, the reference and alternate SNPs appear as in Table 2. The Appendix A records all reference and alternate SNP alleles from each of the gene loci in Table 2 for each host, whereas only SNP alleles relevant to this study are recorded in Figure 4. Figure 5 shows the relationships of sex and each gene’s allele to GCF response strength (Figure 5a–d). Sex was unrelated, but the A allele at *IL1B*-511 did significantly associate with strong GCF response (Figure 5b). In addition, *IL1B*+3954(T) and *IL10*-1082(T) were each a major fraction of weak GCF responses, but neither was significant (Figure 5c,d).

Combinations of alleles are a feature of Figure 4. The genomes of 6 strong GCF responders were homo- or heterozygous for the alternate (A) allele of *IL1B*-511 and homozygous for the reference cytidine (C) allele of *IL1B*+3954 (Figure 4 lanes iii and iv). In addition, the genomes of 3 weak responders possessed the homozygous, reference guanine (G) allele in *IL1B*-511 and the homo- or heterozygous alternate allele (T) in *IL1B*+3954. Thus, SNPs in *IL1B* at −511 and +3954 were exclusively associated with strong or weak GCF exudation in 9 of the 15 hosts. Of the six hosts remaining, five numbered 1, 2, 3, 10 and 12, possessed both *ILB*-511(A), significant for strong GCF response, and *IL1B*+3954(T), significant for weak GCF response. In hosts 2 and 3, this allelic conflict was resolved as a strong response by the homozygous alternate SNP allele (CC) of *IL10*-1082. In host 10, the reference SNP allele (TT) of *IL10*-1082 gave a weak GCF response, similar to the SNP reference allele (AA) of CD14-260 in host 12). 

### 2.2. IL6-1363(T) Acts on Alleles at IL1B+3954 to Reverse the Expected GCF Response (Epistasis)

The above results account for strong or weak GCF exudation from all except hosts 1 and 11 in Figure 4. Host 1 was the 5th host possessing the *IL1B* allelic conflict, and host 11 was an expected strong GCF responder like the six other *IL1B*-511(A)+*IL1B*+3954(CC) hosts. It seems likely that latter’s phenotype was converted to weak by *IL6*-1363(T) in the genome, and that the host 1 phenotype, *IL1B*+3954(T), was converted to strong despite the presence of *IL10*-1082(TT), which should have confirmed a weak response as in host 10. These phenotype reversals exemplify a genetic phenomenon called specific epistasis [42], because it is specific for complete and dominant interaction between *IL6*-1363(T) and the C or T SNP alleles at *IL1B*+3954 [43]. Including these two events made the relationship of *IL1B*+3954(T) significant with respect to GCF response (Figure 5e).

Besides the tooth loss SNP genotype in host 14, a second such genotype was detected in host 11 by *IL6*-1363(T) mediating epistasis of *IL1B*+3954(CC) to weak with IL1A-889(A) in the genome. *IL1B*+3954(T) in the genome not only associates with severe periodontitis [32] but also predicts a weak GCF response in this experimental gingivitis study [9]. We conclude that the two GCF response strengths to biofilm lysine in our prior EG study (Figure 3) are genetically determined strong and weak phenotypic traits. The results are summarized in Table 3 to help clarify the above associations.

### 2.3. IL1B Alleles Alone or with Alleles of IL6, IL10 and CD14 Determine Strong or Weak GCF Response Traits to Biofilm Lysine Content in EG

The strong GCF trait was indicated in hosts 4 through 9 by the alternate *IL1B*-511 alleles (GA or AA) in the genome with the *IL1B*+3954 reference allele (CC). The weak trait was indicated in hosts 13 through 15 with the reference allele (GG) at *IL1B*-511 present with alternate alleles (TC or TT) at *IL1B*+3954. In host 11, a 7th *IL1B*-511(GA) allele paired with *IL1B* +3954(CC) was converted to weak by *IL6*-1363(GT) epistasis of IL1B+3954 (CC) and one of the five hosts possessing the allelic conflict, *IL1B*-511(GA)+*IL1B*+3954(CT), was converted to strong (host 1). Of the remaining four hosts, 2 expressed the strong trait with the homozygous alternate allele, *IL10*-1082(CC) in the genome, and the other two expressed the weak trait with its homozygous reference allele, *IL10*-1082(TT), or a homozygous alternative allele of *CD14*-260(AA) in the genome. In host 3, the presence of *COX2*+8473(AG) specified a potential replacement of *IL10*-1082(CC) for a strong phenotype.

Table 4 shows the various combinations of reference and alternate SNP alleles that predicted a strong or weak GCF response after a week of EG. These were SNP alleles of *IL1B* at −511 and +3954 exclusively, or with the addition of a third SNP allele from a second gene. The third allele was the homozygous reference or alternate SNP of IL10-1082, a homozygous alternate SNP of *CD14*-260, or a heterozygous alternate allele of *IL6*-1363.

## 3. Discussion

### 3.1. Strong and Weak GCF Responses to Biofilm Lysine Are Genetically Determined Phenotypic Traits

A reanalysis of our previously reported EG study [11] revealed a global strong or weak GCF exudation response to the biofilm lysine content (Figure 3, Section 1.3). Different combinations of reference and alternate SNP alleles at *IL1B*-511 and *ILB*+3954, alone, or together with an SNP allele of a second gene, separate all strong from all weak GCF responders (illustrated in the Appendix A). Based on these results, we propose that the strong and weak GCF responses are phenotypic GCF exudation traits that underly the different abundances of bacterial genera detected in gingivally healthy adults before and after EG (Section 1.2). Because the results of this study provide useful outcomes with respect to controlling periodontitis development (Section 3.5), suggestions presented in Table 5 are discussed below.

IL1β remains responsive to inflammasome-mediated IL1α stimulation by biofilms in the gingival crevice throughout life (Section 1.2). Limiting biofilm accumulation, smoking or developing diabetes should protect from periodontitis development. Nevertheless, those who possess the weak trait will likely also have to take more care of their teeth than those who possess the strong trait. Many born with the Group A mixture of IL1B-511 alleles in Table 5 may traverse life with little or no periodontitis, whereas those with the Group B mixture may have to work hard to avoid developing moderate periodontitis by late middle age. These potential consequences of the results in Table 4 and Table 5 are discussed further in Section 3.4 and Section 3.5.

Susceptibility to moderate or severe periodontitis is linked to weak GCF exudation because: (a) Smokers exhibit weak GCF exudation [44] and significantly more periodontitis than non-smokers [1]; (b) Incubating the salivary microbiome from periodontally healthy hosts in a medium resembling GCF for 3 weeks in vitro with only twice weekly supplements results in the appearance of the ‘red’ complex, mimicking weak GCF exudation. This periodontitis-associated mixture of gram-negative bacteria includes *P. gingivalis* [45] and associates with the presence of *IL1B*+3954(T) and moderate or severe periodontitis [32]; (c) Unlike weak GCF exudation, strong GCF exudation expels bacteria efficiently from gingival crevices [46] which assists oral hygiene in retarding ‘red’ complex and periodontitis development.

Of the two hosts possessing *IL6*-1363(T) in their genome, the phenotype of one host with *IL1B*+3954(CC) was converted from strong to weak GCF exudation, and the one possessing the *IL1B*+3954(T) was converted from weak to strong. These phenotypic changes suggest specific epistasis [42], not DNA methylation. Specific epistasis is a gene-allele-mediated reversal of phenotype determined by alleles of another gene [47], a genetic trait unrelated to methylation. Both epistatic genes had the smallest biofilm lysine content of their respective GCF response groups (Appendix A), which correlates with a report of *IL6*-1363(T) being associated with aggressive (severe) periodontitis [38]. The structure of the IL1β protein and its interaction with the RNA around the alternate or reference SNPs at *IL6*-1363 might provide information as to how this form of epistasis operates in periodontitis [48]; Appendix A.

Besides epistasis in two hosts, one of these hosts and 3 others possessed a conflicted genotype, *IL1B*+3954(CC)+*IL1B*-511(GA), here shown as 2 alleles of *IL1B* connected with a plus sign like the two alleles of IL1A and IL1B that make up the *IL1* tooth loss genotype. The conflicted genotype reduces the accuracy with which *IL1B*+3954(T) alone can associate with periodontitis. Almost 300 hosts were required to break through the obstacles of epistasis and conflicted genotypes before a significant relationship of *IL1B*+3954(T) with periodontitis was obtained [32], whereas only 15 hosts were required for this study. We have, therefore, found that the lysine in healthy dental biofilms is the independent variable whereby dental biofilms can best detect strong and weak responses to bacterial agents, pathogen-associated molecular patterns (PAMPs). PAMPs activate a subset of germline-encoded pattern recognition receptors (PRRs) that assemble with other molecules into inflammasomes that activate *IL1A* and *IL1B* [16]. The result is a strong or weak GCF exudation trait that determines periodontal disease susceptibility (Table 4 and Table 5), depending on which SNP variants of *IL1B* at −511 and +3954 are present in the genome. Table 4 and Table 5 also indicate that the SNP allele at *IL1A*-889 is a secondary factor, and not critical for determining GCF exudation trait.

### 3.2. Greater Expression of the Selected Genes Promotes the Weak GCF Trait

The complexity of interactions required to induce an innate immune response requires optimal stimulation of the appropriate genes to induce acquired immunity, antibody and thymus (T) cell-mediated immunity which protects from bacterial invasion of the gingival crevice [16]. We suggest that sub-optimal expression of these genes can compromise host survival, whereas overproduction of an interleukin or other encoded protein merely over-stimulates systemic inflammation that will at least let a host survive longer with an infection despite a lack of acquired immunity. Of the genes tested, the alternative alleles enhanced expression of their encoded protein except for *IL1B*-511, *IL10*-1082, and CD14-260 in which this function is present in their homozygotic reference (GG) or (TT) alleles (rs16944 and rs1800896). Unlike the other 3 genes, it is specifically the alternate SNPs, (A) allele of *IL1B*-511 and (C) allele of IL10-1082, that promote strong GCF exudation, efficient bacterial removal from the crevice and mild or no periodontitis (Table 1).

A related question is whether natural protection from periodontitis can be maintained long enough to permit an EG test in hosts who have reached their early 60s. A 3-week EG experiment used five young hosts aged 20–22 years and five old hosts aged 61 to 65 years, all Caucasians [49]. PI, GI, and GCF volumes were significantly greater in the old group. IL1α levels in both groups had increased significantly by day 21 and returned to baseline a week after the 3-weeks of EG had ended (day 28). On the other hand, IL1β levels increased significantly only in the older group before falling back to starting levels by day 28. One possibility is that the *IL1B*-511(A)+*IL1B+*3954(CC) genotype reported in Table 4 can, with environmental help, give almost lifetime protection. This combination could be more protective than either the undetected *IL1B*-511(GG)+*IL1B+*3954(CC) genotype in which *IL1B*-511(GG) produces 25% more IL1β than its alternative (A) allele [38], or the genotype conflict pair, *IL1B*+3954(T)+*IL1B*-511(A). Conversely, the concurrent presence of all three ‘red’ complex periodontal pathogens and IL1B+3954(T) associated with the greatest IL-1β expression in GCF from periodontitis sites [50]. 

Chronic periodontitis rarely appears before age 30 (Table 1), about 15 years after gingivitis occurs (Section 1.1). The long incubation time lets the crevices of weak GCF responders incubate the ‘red’ complex containing *P. gingivalis*. Within a group of 34 middle-aged, non-smoking US Caucasian patients with periodontitis, *P. gingivalis* colonization in the deepest periodontal pockets reached almost 50% of total bacteria in 44% of the patients compared with only 12% in the remaining 56% [51]. This fraction of high *P. gingivalis* colonization in a middle-aged population with moderate periodontitis corresponds to the fraction of weak GCF responders in our original EG study of 16 patients (44% from Figure 3).

### 3.3. How Weak GCF Exudation Promotes P. gingivalis Infection and Periodontitis

Differences in innate immunity determine how cancers are controlled by the host [52]. This information is adapted in Figure 6 to summarize the results of the innate immune response mediated by alleles of *IL1B*, *IL6*, *IL10* or *CD14*, to control or advance periodontitis and its comorbidities. After salivary bacteria adhere to teeth and extend into previously healthy gingival crevices during EG for a week [15], inflammasomes are activated and express IL1α and IL1β (Section 1.2). Both these interleukins secrete proinflammatory cytokines which enter the bloodstream and upregulate polymorphonuclear (PMN) cell production in the bone marrow [53]. Increased mobilization and priming make these PMNs spill into the bloodstream from which they enter the gingiva and cross through the JE into gingival crevices [54].

If PMNs become fully activated (‘hot’), they digest bacterial products, and protect the host by inducing acquired immunity (Section 1.2). If PMNs are less activated ‘cold’, the reduced activation of acquired immunity produces weak GCF exudation and ‘red’ complex development [55]. Our weak GCF responder trait is associated with the presence of immune suppressive ‘cold’ PMNs. The association of ‘cold’ PMNs with weak GCF trait in developing gingivitis lets *P. gingivalis* and other bacteria from the ‘red’ microbial complex eventually appear in periodontal pockets, where they cause periodontitis and subsequently spread systemically to promote the many age-related comorbidities related to periodontitis [6].

### 3.4. Weak GCF Trait, Periodontitis, CVD and Dementia

*P. gingivalis* expresses outer membrane vesicles (OMVs) that spread from periodontitis pockets throughout the body to cause as many as 20 different diseases [6]. This organism is unique to periodontal infections and may promote CVD by the OMVs attaching to the inner wall of blood vessels [56] where their contents may alter the stability of atherosclerotic lesions called plaques [6]. In CVD, cholesterol from lipoproteins in the blood accumulates on the inner wall of blood vessels as atherosclerotic lesions. The proteases and other enzymes in OMVs from *P. gingivalis* infection may destabilize the atherosclerotic plaques and make them more likely to detach, spread and become trapped in small arteries or veins [57]. The plaque fragments could also spread to the brain where they might cause cerebral small vessel disease (SVD) which accounts for almost half of all dementias. SVD can be confused with Alzheimer’s disease unless separated by the appearance of the brain after magnetic resonance imaging [57].

Statins modify this process by inhibiting the enzyme that synthesizes cholesterol [58], significantly retarding clotting, strokes, and dementia [59]. Guo et al. [5] performed a meta-analysis that revealed an association between periodontitis and cognitive impairment. They reported that moderate or severe periodontitis was a risk factor for dementia. We, therefore, suggest that our test for weak GCF trait could predict susceptibility to infection by *P. gingivalis* causing moderate and severe periodontitis [60], and its comorbidities such as SVD, Alzheimer’s, disease, and other dementias. A start would be to compare the ratio of weak responders who are over age 65 and have dementia with those of similar age who do not have dementia.

Genetic studies of alleles based on the co-existence of periodontitis and systemic diseases are scarce, but one report utilizes the observation that the presence of the reference allele of *IL1B*-511(GG) displays greater protein expression and greater cytokine activation than its alternate allele, AA, or GA [36]. When 135 Chinese patients with Alzheimer’s disease were classified into 108 G, and 27 non-G *IL1B*-511 individuals, that exclusively differed in the entorhinal-cingulum axis. *IL1B*-511(GA) polymorphisms modulated the structural covariance strength on the anterior brain and entorhinal-interconnected networks independently of white matter tract integrity. The *IL1B*-511 GG and GA groups were weaker than AA carriers in covariance strength, and the different network clusters associated with each *IL1B*-511 GA genotype could predict the cognitive test result [61].

The prediction of a cognitive test result applies to Chinese patients and may not apply to Hungarian patients in our study. One of our current proposals is to determine whether patients with Alzheimer’s and/or other cognitively impaired diseases in Hungary are more likely to possess the weak GCF trait and greater *P. gingivalis* infection than similar individuals from the same population who did not have dementia. How that result would compare with the role of *IL1B*-511(GA) alone in being able to predict a cognitive test result is uncertain.

### 3.5. Using GCF Phenotypic Traits to Develop New Methods to Prevent or Control Periodontitis

The primary procedure for controlling periodontal disease is oral hygiene, self-administered twice daily by toothbrushing and twice yearly by professional scaling, root-planing and prophylaxis. As noted in Section 1.4, the tooth loss genotype, *IL1A*-889(A)+*IL1B*+3954(T) occurs at a frequency of 68% in hosts presenting with at least two of three periodontitis risk factors, smoking, diabetes, or teeth cleaned professionally twice or less per year [8]. We have greatly enlarged the phenotypic and genetic information to potentially detect about 40% of the population who are at risk for developing moderate or severe periodontitis (Table 1). For example, tooth loss data suggests that twice-yearly tooth cleaning may not provide adequate protection from disease in hosts identified to have the weak GCF trait. Testing for strong and weak GCF trait phenotypes could contribute to better clinical prevention of periodontal inflammation including dental implant failure [62] and their comorbidities.

## 4. Materials and Methods

### 4.1. Subject Selection and Follow-up

The starting point of this article was a preliminary study conducted at Semmelweis University [26] and concluded with an EG study published in the Journal of Periodontology in 2012 [11]. That study used the biofilm lysine concentration (µL lysine/g biofilm) as the independent variable instead of PI, and GCF exudation (µL/min) as the dependent variable (Appendix A). The Appendix A, reproduces the Methods section of this EG study [11], and the setting, location, periods of recruitment, exposure, follow-up, and data collection from 16 hosts.

DNA for the present study was obtained from all the participants in that EG study, except a female participant who was abroad. There was no additional recruitment, exposure, follow-up, or repeated data collection. The Appointments for cheek scrapings were made and the scrapings were stored at −70 °C in separate containers until the DNA was extracted, purified and tested for the presence or absence of alleles from each of the genes listed in Table 2.

### 4.2. Measurement of Alleles of Genes Associated with Periodontitis

We used the TaqMan assay described by ThermoFisher Scientific (Waltham, MA, USA) to genotype the SNP alleles at each locus in each of the 7 genes listed in Table 2 (Section 2.1). Each assay kit contained two primers for amplifying the sequence of interest and two allele-specific and differently labeled TaqMan minor groove binder (MGB) probes for allele detection. One end of each allele-specific MGB primer had a covalently attached fluorescent reporter molecule (FAM or a VIC), and the other 3′ ends had an attached fluorescence quencher. During the PCR amplification step the allele-specific probe was perfectly complementary to the SNP allele and it hybridized to the target DNA segment. Fluorescence measurements during genomic DNA amplification in the presence of both primers were graphed over time. From this graph, the fluorescence reporter molecule type indicated which allele was present in the host’s genomic DNA (https://www.thermofisher.com/lu/en/home/life-science/pcr/real-time-pcr/real-time-pcr-assays/snp-genotyping-taqman-assays.html, accessed on 3 April 2023).

### 4.3. Statistical Analyses of GCF Responses to Biofilm Lysine after a Week of EG

Strong and weak responses used biofilm lysine concentration (µg/mg biofilm) as the independent variable and GCF response (µL/min) as the outcome. A quadratic polynomial regression model (JMP Pro 13.1 computer program, SAS Institute Inc., Cary, NC, USA) was employed. The significance of the GCF response (main effect) and its interactions with lysine content and lysine content-squared (independent variables) were determined using an F-ratio test with 3 degrees of freedom to compare the sum of squares explained by the two independent terms (Lys and Lys-squared) relative to the sum of squares error (legend to Figure 4). Differentiation between strong and weak GCF response groups by sex and by the alleles of selected genes was determined by Fisher’s exact statistic (https://www.socscistatistics.com/,accessed on 3 June 2022).

## 5. Patents

Devices, kits, and methods for determining increased susceptibility to and treatment and prevention of periodontitis, Alzheimer’s disease, and other conditions. PCT/US22/33045. Received 10 June 2022, Inventor, Martin Levine.

## Figures and Tables

**Figure 1 ijms-24-14249-f001:**
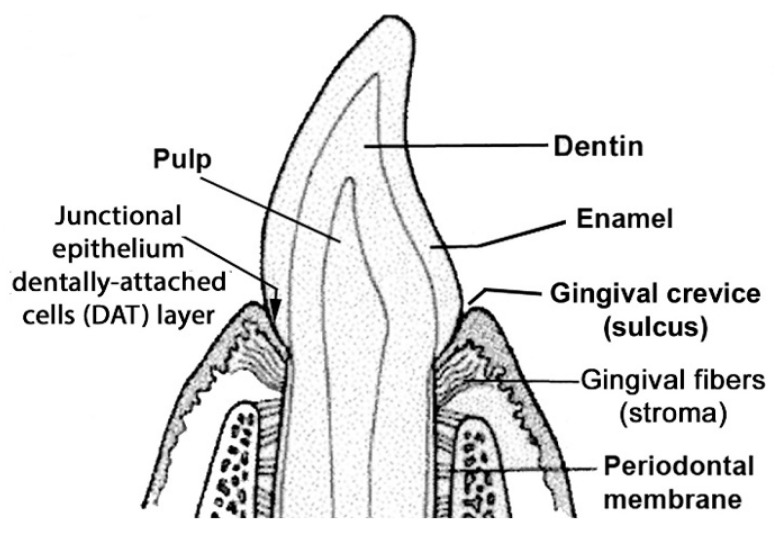
The healthy junctional epithelium (JE). Arrowhead on the left indicates the dentally attached (DAT) basal cells, and on the right, those attached above the stroma.

**Figure 3 ijms-24-14249-f003:**
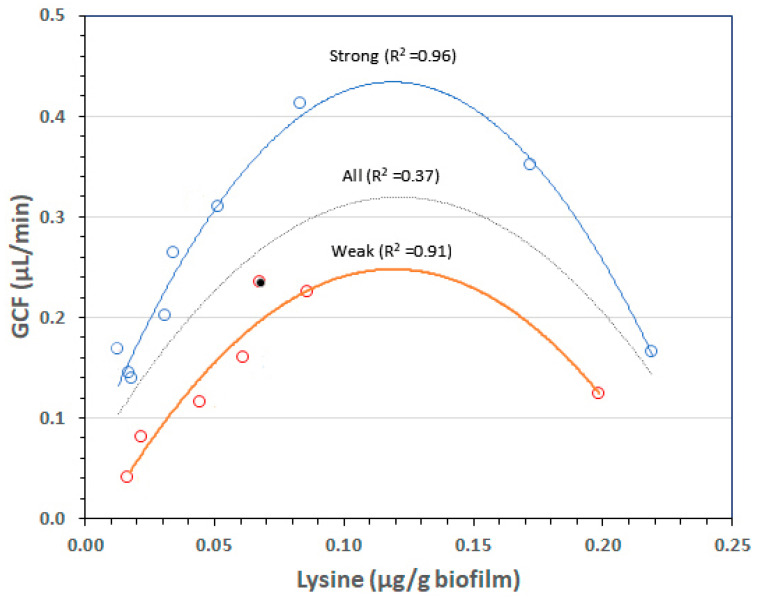
Dual relationship of biofilm lysine content to gingival crevicular fluid (GCF) after oral hygiene restriction for a week. The central, gray curved line indicates the single quadratic relationship shown in Appendix A. A much stronger fit of parallel strong (blue) and weak (red) GCF response curves was highly significant (F Ratio 41.77, *p* < 0.0001). Interactions with biofilm lysine and lysine-squared contents were not significant. A strong GCF response to the lysine content of the biofilm (blue line) was about twice that of the weak response (red line). This figure as a poster at the EuroPerio9 conference in Amsterdam, in June 2018 and within an oral presentation at the IADR-PER meeting in July 2018.

**Figure 4 ijms-24-14249-f004:**
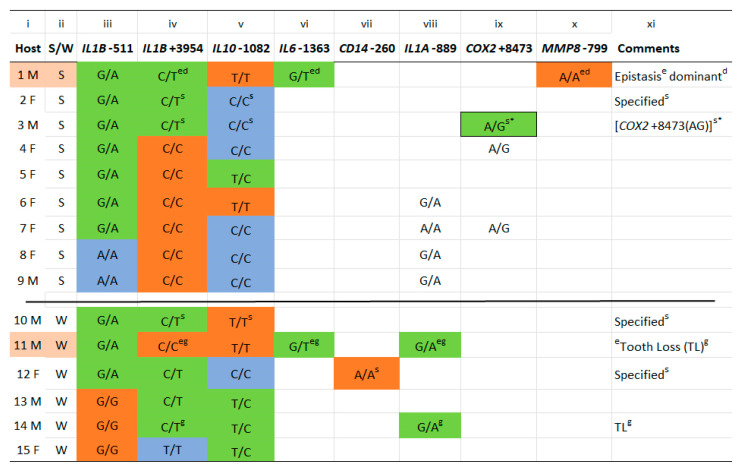
Alleles of genes that identified strong and weak GCF exudation. Gene names are abbreviated and italicized by convention. Plus/minus numbers immediately following the gene name indicate the number of nucleotides upstream or downstream from the polypeptide-initiating nucleotide that encodes the translation start site. The reference allele is shown first, and if the SNP alternate allele is shown after the slash (heterozygous), the box background is colored green. Otherwise, if both copies contain the reference SNP (homozygous) and the gene is translated from the forward (5′) strand, the reference gene is colored red and the alternate homozygous gene is colored blue (lanes iii and iv). If the reference SNP is translated from the complementary (3′ strand), the homozygous reference box is colored blue, and the homozygous alternate SNP is colored red (lane v). SNPs in boxes with a white (uncolored) background indicate a third, SNP allele specifying (s) a strong or weak GCF response, or the tooth loss genotype (g) in one or more hosts. Unfilled boxes indicate alleles unrelated to the study. Comments: Epistasis^e^ indicates the presence of the alternate T SNP at *IL6*-1363 in hosts 1 and 11, and its dominance^d^ in host 1. Specified^s^ indicate the allele of a second gene required to differentiate hosts possessing both *IL1B*-511(GA) and *IL1B*+3954(CT). Square brackets and the symbol * identify host 3 in whom an alternative third gene’s allele may specify the same GCF response; Genotype^g^ indicates the presence of the tooth loss genotype, IL1A-889(A)+IL1B+3954(T) [8]. In host 11, epistasis of *IL1B*+3954(CC)^e^ created a second tooth loss genotype^g^.

**Figure 5 ijms-24-14249-f005:**
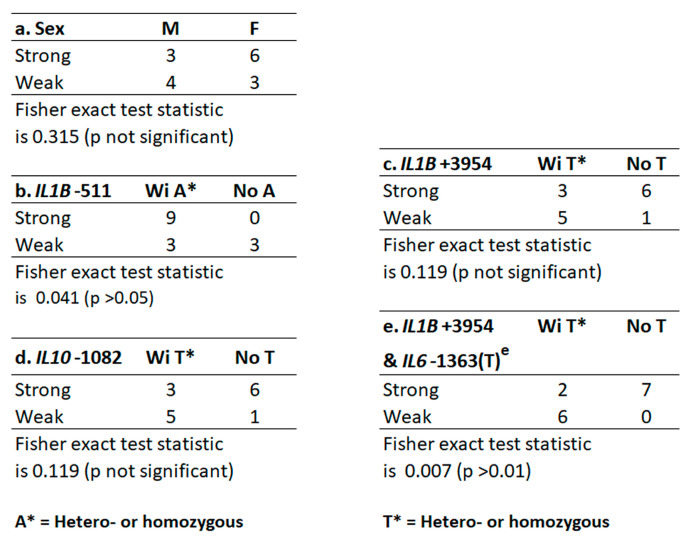
Independent Fisher tests for GCF strong or weak exudation trait. (**a**). Sex by GCF; (**b**). *IL1B* allele at −511; (**c**). *IL1B* allele at +3954; (**d**). *IL10* allele at −1082; (**e**). *IL1B*+3954 with epistatic *IL6*-1363(T).

**Figure 6 ijms-24-14249-f006:**
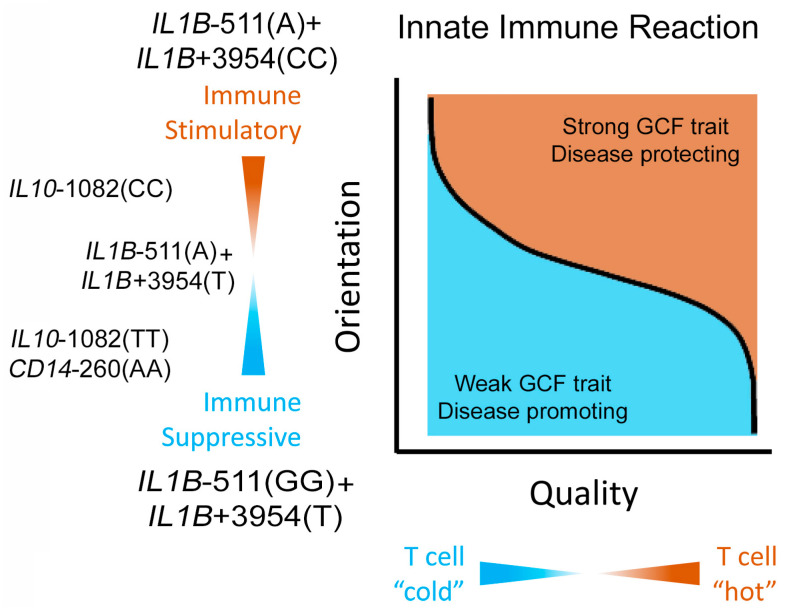
The quality and orientation of the innate immune reaction to salivary bacteria in the gingival crevice define periodontitis control. The orientation of the immune response refers to inflammation inclined towards supporting or inhibiting periodontitis. Our results suggest that alleles of *IL1B* at −511 and +3954 determine strong or weak PMN infiltrates whose cytokines mediate a corresponding GCF response trait. The quality of that response refers to the ability of the induced inflammation to activate acquired immunity with dendritic and thymus (T)-cells. The *IL1B* alleles activate strong or weak PMNs that, together with an allele of *IL6* (Figure 4) activate “hot” or “cold” T-cells infiltration. A weak GCF trait indicates weak subclinical inflammation, and a ‘cold’ T-cells response promotes periodontitis and its comorbidities independently of environmental factors such as oral hygiene or smoking. The conflicted genotype (on the left middle side of the graph) has a smaller letter size to indicate that it is uncommon compared to the 9 individuals whose actions require only the 2 alleles of IL1B. Adapted from Chang and Beatty [52]. The interplay between innate and adaptive immunity in cancer shapes the productivity of cancer immunosurveillance. J. Leukoc. Biol. 2020, 108, 363–376. (Creative Commons Copyright permission from Publisher, Oxford University Press).

**Table 1 ijms-24-14249-t001:** Prevalence of periodontitis in US aged 30 years and above. Original table from text of reference [1].

Severity	Prevalence
No disease	52.80%
Mild	8.70%
Moderate	30.00%
Severe	8.50%
Total	100.00%

**Table 2 ijms-24-14249-t002:** Selected genes associated with periodontitis. ^a^ Gene name is italicized, and the SNP allelic site location is indicated by the number of nucleotides before (-) or after (+) the translation start site. Reference and alternate SNP nucleotides are bracketed with the reference SNP first. ^b^ Reference SNP cluster ID number. ^c^ Minor allele frequency (MAF) validated in a large European population. ^d^ Co-dominant alleles are defined as present at more than 40% of individuals in a large population.

Gene, Site, (SNP) ^a^	rs Number ^b^	MAFα Euro ^c^
*IL1A*-889(G/A)	rs1800587	0.285
*IL1B*+3954(C/T)	rs1143634	0.237
*IL1B*-511(G/A)	rs16944	0.335
*IL6*-1363(G/T)	rs2069827	0.085
*IL10*-1082(T/C)	rs1800896	0.527 ^d^
*CD14*-260(A/G)	rs2569190	0.485 ^d^
*COX2*+8473(G/A)	rs5275	0.337
*MMP8*-799(G/A)	rs11225395	0.464 ^d^

**Table 3 ijms-24-14249-t003:** Summary of genotype alleles that detected strong or weak GCF traits. Asterisks here indicate the 5 hosts possessing the conflict allele, *IL1B*+3954(T)+*IL1B*-511(A). Explanations for each superscript letter including the superscript asterisk were previously indicated by an underline in the legend to Figure 4.

Host	Sex	Trait	Genotype		2nd Gene	Comments
1	M	S	**IL1B+3954(CT)^e^*	*IL1B*-511(GA)	*IL6*-1363GT^e^	Epistasis^e^
2	F	S	**IL1B*+3954(CT)	*IL1B*-511(GA)	*IL10*-1082(CC)^s^	Specified^s^
3	M	S	**IL1B*+3954(CT)	*IL1B*-511(GA)	*IL10*-1082(CC)^s^	[*COX2*+8473(AG)]^s*^
4	F	S	*IL1B*+3954(CC)	*IL1B*-511(GA)		
5	F	S	*IL1B*+3954(CC)	*IL1B*-511(GA)		
6	F	S	*IL1B*+3954(CC)	*IL1B*-511(GA)		
7	F	S	*IL1B*+3954(CC)	*IL1B*-511(GA)		
8	F	S	*IL1B*+3954(CC)	*IL1B*-511(GA)		
9	M	S	*IL1B*+3954(CC)	*IL1B*-511(GA)		
10	M	W	**IL1B*+3954(CT)	*IL1B*-511(GA)	*IL10*-1082(TT)^s^	
11	M	W	*IL1B*+3954(CC)^eg^	*IL1B*-511(GA)	*IL6*-1363(GT)^eg^	*^e^*[*IL1A*-889(GA)]^g^
12	F	W	**IL1B*+3954(CT)^s^	*IL1B*-511(GA)	*CD14*-260(AA)^s^	Specified^s^
13	M	W	*IL1B*+3954(CT)	*IL1B*-511(GG)		
14	M	W	*IL1B*+3954(CT)^g^	*IL1B*-511(GG)	*IL1A*-889(GA)^g^	Genotype^g^
15	F	W	*IL1B*+3954(TT)	*IL1B*-511(GG)		

**Table 4 ijms-24-14249-t004:** Summary of IL1B gene combinations, GCF trait, and periodontitis. ^1^ GCF trait: S = Strong; W = Weak. ^2^ Periodontitis: Protected = No disease or Mild; Susceptible = Moderate or Severe. ^3^ 2nd gene determines periodontitis susceptibility. Superscripts ^e^ and ^s^ are listed in Figure 4 and Table 3.

# of Hosts		IL1B+3954	IL1B-511	Predicted ^1^	Periodontitis ^2^
7	47%	*IL1B*+3954(CC)	*IL1B*-511(GA)	6S+1W^e^	Protected
3	20%	*IL1B*+3954(CT)	*IL1B*-511(GG)	3W	Susceptible
5	33%	*IL1B*+3954(CT)	*IL1B*-511(GA)	2S^s^+1S^e^+2W^s^	2nd Gene ^3^
0	0	*IL1B*+3954(CC)	*IL1B*-511(GG)	Not present	Not available

**Table 5 ijms-24-14249-t005:** Determination of GCF trait from SNP allelic genotypes.

Groups	Strong or weak GCF Traits from Allelic Genotypes
A.	*IL1B*+3954(CC)+*IL1B*-511(GA) = strong (6 hosts)
B.	*IL1B*+3954(CT)+*IL1B*-511(GG) = weak (3 hosts)
B1.	*IL1B*+3954(T)+*IL1B*-511(GG)+*IL1A*-889(A) = weak (tooth loss gene)
C.	*IL6*-1363(T) reverses phenotype, specific epistasis (2 hosts)
C1.	*IL1B*+3954(CC)+*IL1B*-511(GA) = strong
C2.	*IL1B*+3954(CT)+*IL1B*-511(GA) = weak
C2.1.	*IL1B*+3954(CC)+*IL1B*-511(GA)+*IL6*-1363(T)+*IL1A*-889(GA) = weak
	(Epistasis with tooth loss gene)
D.	*IL1B*+3954(CT) and *IL1B*-511(GA) conflicted, but with:
D1.	Presence of *IL10*-1082(CC) or *COX2*+8473(AG) = strong (2 hosts).
D2.	Presence of *IL10*-1082(TT) or *CD14*-260(AA) = weak (2 hosts).
E.	*IL1B*+3954(CC)+*IL1B*-511(GG) = GCF trait uninterpretable (no hosts)

## Data Availability

The authors agree to share their research data, none of which is related to our patent. The patent is exclusively for methods that enable alleles of all the genes reported here to be tested more rapidly than currently.

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
