# Peer review of "Genetic Control of GCF Exudation: Innate Immunity Genes and Periodontitis Susceptibility"

_ijms, 2023, doi:10.3390/ijms241814249_

Round 1
Reviewer 1 Report
Topic and presented results are of great interest for readers.
However, the presentation would be revised.
Introduction and discussion made up the majority of the present paper.
Futheremore, there are multiple repetion of same information.
ex) Fig3 & 4 overlap information.
"2.4. summary of the results" is not needed.
The authors should present background and results of the study clearly and consisely.
Author Response
Response to Reviewer1:
Thank you for your interest. We have revised the presentation.
- The text of the article is greatly changed, and its length is reduced by a page to remove repetition.
- The summary of the results is also deleted along with Figure 3, which we moved to the supplement, along with its legend and details of the experimental gingivitis method from Reference 11.
- We have made major changes to the Discussion as indicated in our response to reviewer 2.
Reviewer 2 Report
The article is interesting and shows new information about SNPs and periodontitis. However, some points need to be improved. Thus, the study needs a major revision.
Title:
The title is too long and is not fully associated with what the authors have done. For example: the authors do not have data on comorbidities. The title needs to be more concise and not include hypotheses.
Introduction:
- Authors must include the biological effect of each of the SNPs.
- Table 2: Authors must include the source used to obtain the MAF value for Europeans.
- Materials and methods:
- Line 472: The authors insist on highlighting Dr. Lohinai, but this is not a practice in scientific articles. The study must be cited as it is usually done, that is, Lohinai et al. (year).
- Lines 484-486: The sentence is truncated.
- Section 4.1: Even if the present study is a continuation of a previous study (Lohinai et al. 2012), it is important that the authors include a summary of the studied population, inclusion and exclusion criteria, etc.
Discussion:
- Authors should discuss the data obtained by associating the biological effect of SNPs. Are these SNPs associated with changes in expression? on cytokine activity?
- Line 348: Periodontal disease is multifactorial, and in this sense few polymorphisms may not explain its origin. Other SNPs and methylation profile in the genes studied by the authors must be considered and could explain some data obtained by the authors. For example, Ishida et al. (2012) found an association between the IL6 methylation profile and IL6 levels in blood tissue from patients with periodontitis (DOI: 10.1902/jop.2011.110356). Kobayashi et al (2016) also found an association between the IL6 methylation profile in patients with periodontitis (10.1016/j.archoralbio.2016.05.018). See also data on IL1-beta (DOI: 10.1016/j.archoralbio.2023.105694).
- Line 382: Quote the Chinese reference.
- Line 464-469: Authors should research whether there are studies showing association of polymorphisms in IL1Beta and IL6 and dementia.
Minor editing of English language required
Author Response
Response to reviewer 2
Thank you for the detailed comments.
- The title of the article is shortened to “Genetic control of GCF exudation: innate immunity genes and periodontitis susceptibility.”
- Selected SNP alleles were all previously associated with periodontitis.
- The source, sequence and details of each SNP allele are from the NCBI reference (rs) number provided in Table 2. Additional details are given in the legend to Table 2, and on lines 135-137 of the text. To search in the NCBI database, each number must be preceded by the letters rs, and the initial search should be across all databases.
- Highlighting Dr. Lohinai is removed from the revised article, see line 367.
- Truncated sentence corrected in revised article, lines 378-381.
- A summary of the studied population, the inclusion and exclusion criteria, etc. are added to the Supplement, Item 1.
- Biological effect of SNPs See section 3.3. of the revised discussion. We have briefly alluded to innate immunity and how that activates IL1 to interact with neutrophils. The response to IL1B-induced cytokines is inflammation until foreign agents and tissue damage are removed. Our results suggest that the strength of that innate response depends on 2 SNP variant alleles of IL1B, alone or with an allele in a third gene. See details in revised Section 3.3. and Figure 4.
- Based on Table 6, it is possible that the variant T allele of IL1B+3954 is less methylated than its common homozygous (C) allele, and that the homozygous G allele of IL1B-511 is less methylated than the variant A allele. We are uncertain what this would mean for our study or its conclusions. See Reference [1] below.
- Section moved and Chinese reference clarified in the revised article, lines 78 - 81.
- Reference [2]is a study of IL1B-511 AA and GG alleles, but it is not clear how results of that study relate to our study. Unlike reference 2, we are not looking at IL1B-511 alleles by themselves, but rather in combination with alleles of IL1B+3954. Moreover, they looked only at at homozygous C or T alleles (today called G or A alleles due to a new rule about direction of translation). We found that these homozygous common or variant alleles are uncommon in our EG study; 10 of our 15 hosts were heterozygous (A and G). Currently, we are preparing a proposal to determine whether we can compare the frequency of the weak GCF trait in Alzheimer’s patients with a similar group of non-Alzheimer’s control patients.
References
- Ali, M.M.; Naquiallah, D.; Qureshi, M.; Mirza, M.I.; Hassan, C.; Masrur, M.; Bianco, F.M.; Frederick, P.; Cristoforo, G.P.; Gangemi, A.; et al. DNA methylation profile of genes involved in inflammation and autoimmunity correlates with vascular function in morbidly obese adults. Epigenetics 2022, 17, 93-109, doi:10.1080/15592294.2021.1876285.
- Huang, C.W.; Hsu, S.W.; Tsai, S.J.; Chen, N.C.; Liu, M.E.; Lee, C.C.; Huang, S.H.; Chang, W.N.; Chang, Y.T.; Tsai, W.C.; et al. Genetic effect of interleukin-1 beta (C-511T) polymorphism on the structural covariance network and white matter integrity in Alzheimer's disease. J Neuroinflammation 2017, 14, 12, doi:https://doi.org/.
Round 2
Reviewer 2 Report
The authors made some of the suggested modifications and others were not found in the text. Since the authors did not use a very common practice in response to reviewers, that is, highlighting the changes in color in the text, it is difficult to follow. The manuscript needs minor revision.
- Authors: The title of the article is shortened to “Genetic control of GCF exudation: innate immunity genes and periodontitis susceptibility.” Reviewer: The title is better.
- Authors: Selected SNP alleles were all previously associated with periodontitis. Reviewer: The biological effect of SNPs were not included (or were not highlighted in color in the text). Please insert (or highlight). Are these polymorphisms associated with changes in expression? Changes in protein activity? Please include and highlight in color in the text.
- Authors: The source, sequence and details of each SNP allele are from the NCBI reference (rs) number provided in Table 2. Additional details are given in the legend to Table 2, and on lines 135-137 of the text. To search in the NCBI database, each number must be preceded by the letters rs, and the initial search should be across all databases. Reviewer: NCBI is a database that supports other databases of genomic data, for example: 1000 Genomes, HapMap and etc. Which of these databases are the authors referring to?
- Authors: Highlighting Dr. Lohinai is removed from the revised article, see line 367. Reviewer: In the version I received line 367 does not match what the authors are saying. Please highlight in color in the text.
- Authors: Truncated sentence corrected in revised article, lines 378-381. Reviewer: In the version I received lines 378-381 does not match what the authors are saying. Please highlight in color in the text.
- Authors: A summary of the studied population, the inclusion and exclusion criteria, etc. are added to the Supplement, Item 1. Reviewer: The authors included the requested information.
- Authors: Biological effect of SNPs See section 3.3. of the revised discussion. We have briefly alluded to innate immunity and how that activates IL1 to interact with neutrophils. The response to IL1B-induced cytokines is inflammation until foreign agents and tissue damage are removed. Our results suggest that the strength of that innate response depends on 2 SNP variant alleles of IL1B, alone or with an allele in a third gene. See details in revised Section 3.3. and Figure 4. Reviewer: The authors should discuss the results with data on the biological effect of the SNPs, already described in the literature: is the base change associated with alterations in the expression? on protein activity? Please insert and highlight in color in the text.
- Authors: Based on Table 6, it is possible that the variant T allele of IL1B+3954 is less methylated than its common homozygous (C) allele, and that the homozygous G allele of IL1B-511 is less methylated than the variant A allele. We are uncertain what this would mean for our study or its conclusions. See Reference [1] below. Reviewer: The reference that the authors show [1] does not support the authors' conclusion about polymorphisms and methylation. In fact, the article mentioned doesn't even address polymorphisms. I don't know where the authors got the conclusion that the mentioned alleles could be more or less methylated, even because DNA methylation occurs mainly in CG dinucleotides. DNA methylation has effects on gene expression and could explain why such genotypes do not correspond to the same phenotype when compared with individuals with the same genotype. As periodontitis is a multifactorial disease, both polymorphisms and DNA methylation may be associated. Example: an individual has a polymorphism associated with decreased gene expression in a certain gene and concomitantly has DNA hypomethylation (which is associated with increased gene expression) in that same gene. See that these two mechanisms have antagonistic effects and can explain why the analysis of a genotype alone does not fully explain the phenotype.
- Authors: Section moved and Chinese reference clarified in the revised article, lines 78 - 81. Reviewer: The reference has been included
- Authors: Reference [2]is a study of IL1B-511 AA and GG alleles, but it is not clear how results of that study relate to our study. Unlike reference 2, we are not looking at IL1B-511 alleles by themselves, but rather in combination with alleles of IL1B+3954. Moreover, they looked only at at homozygous C or T alleles (today called G or A alleles due to a new rule about direction of translation). We found that these homozygous common or variant alleles are uncommon in our EG study; 10 of our 15 hosts were heterozygous (A and G). Currently, we are preparing a proposal to determine whether we can compare the frequency of the weak GCF trait in Alzheimer’s patients with a similar group of non-Alzheimer’s control patients. Reviewer: Since the authors included an item on: How periodontitis relates to its comorbidities and emphasize dementia, a discussion at the level of polymorphisms associated with the two diseases is expected. Genetic studies based on the co-existence of periodontitis and systemic diseases are still scarce and need to be further explored. The proposal that the authors are preparing is interesting and could come with a perspective of the present work.
Author Response
Open Review
(x) I would not like to sign my review report
( ) I would like to sign my review report
Quality of English Language
(x) I am not qualified to assess the quality of English in this paper
( ) English very difficult to understand/incomprehensible
( ) Extensive editing of English language required
( ) Moderate editing of English language required
( ) Minor editing of English language required
( ) English language fine. No issues detected
|
Yes |
Can be improved |
Must be improved |
Not applicable |
|
|
Does the introduction provide sufficient background and include all relevant references? |
( ) |
(x) |
( ) |
( ) |
|
Are all the cited references relevant to the research? |
(x) |
( ) |
( ) |
( ) |
|
Is the research design appropriate? |
(x) |
( ) |
( ) |
( ) |
|
Are the methods adequately described? |
(x) |
( ) |
( ) |
( ) |
|
Are the results clearly presented? |
(x) |
( ) |
( ) |
( ) |
|
Are the conclusions supported by the results? |
( ) |
(x) |
( ) |
( ) |
Comments and Suggestions for Authors
The authors made some of the suggested modifications and others were not found in the text. Since the authors did not use a very common practice in response to reviewers, that is, highlighting the changes in color in the text, it is difficult to follow. The manuscript needs minor revision.
- Authors: The title of the article is shortened to “Genetic control of GCF exudation: innate immunity genes and periodontitis susceptibility.” Reviewer: The title is better. Thank you,
- Authors: Selected SNP alleles were all previously associated with periodontitis. Reviewer: The biological effect of SNPs were not included (or were not highlighted in color in the text). Please insert (or highlight). Are these polymorphisms associated with changes in expression? in protein activity? Please include and highlight in color in the text. Although the genes and alleles in Table 2 were previously referenced as being associated with periodontal disease, the annotations to Table 2 and the surrounding text are revised and highlighted to emphasize the connection (Lines 146 – 173).
- Authors: The source, sequence and details of each SNP allele are from the NCBI reference (rs) number provided in Table 2. Additional details are given in the legend to Table 2, and on lines 135-137 of the text. To search in the NCBI database, each number must be preceded by the letters rs, and the initial search should be across all databases. Reviewer: NCBI is a database that supports other databases of genomic data, for example: 1000 Genomes, HapMap and etc. Which of these databases are the authors referring to? See information on lines 134 – 137 of the text.
- Authors: Highlighting Dr. Lohinai is removed from the revised article, see line 367. Reviewer: In the version I received line 367 does not match what the authors are saying. Please highlight in color in the text. The text where Dr. Lohinai was previously mentioned is removed. See highlighted text in Section 4.1. lines 432 – 434.
- Authors: Truncated sentence corrected in revised article, lines 378-381. Reviewer: In the version I received lines 378-381 does not match what the authors are saying. Please highlight in color in the text. The originally truncated sentence is completed and highlighted in green (lines 376 – 377) within a yellow heighted region (lines 374 – 381) which introduces a publication describing a substantial difference in P. gingivalis colonization in patients with moderate periodontitis corresponding to the ratio of weak to strong GCF traits in this article. The result implies that weak responders develop more P. gingivalis and more disease than strong responders. The further implications of this finding are considered in the remainder of section 3.4, i.e., this section of the manuscript. We have also added a new section (3.5., lines 421 – 432) in which we consider
- Authors: A summary of the studied population, the inclusion and exclusion criteria, etc. are added to the Supplement, Item 1. Reviewer: The authors included the requested information.
- Authors: Biological effect of SNPs See section 3.3. of the revised discussion. We have briefly alluded to innate immunity and how that activates IL1 to interact with neutrophils. The response to IL1B-induced cytokines is inflammation until foreign agents and tissue damage are removed. Our results suggest that the strength of that innate response depends on 2 SNP variant alleles of IL1B, alone or with an allele in a third gene. See details in revised Section 3.3. and Figure 4. Reviewer: The authors should discuss the results with data on the biological effect of the SNPs, already described in the literature: is the base change associated with alterations in the expression? on protein activity? Please insert and highlight in color in the text. The new discussion is section, 3.2. It is highlighted on lines 299 – 322.
- Authors: Based on Table 6, it is possible that the variant T allele of IL1B+3954 is less methylated than its common homozygous (C) allele, and that the homozygous G allele of IL1B-511 is less methylated than the variant A allele. We are uncertain what this would mean for our study or its conclusions. See Reference [1] below. Reviewer: The reference that the authors show [1] does not support the authors' conclusion about polymorphisms and methylation. In fact, the article mentioned doesn't even address polymorphisms. I don't know where the authors got the conclusion that the mentioned alleles could be more or less methylated, even because DNA methylation occurs mainly in CG dinucleotides. DNA methylation has effects on gene expression and could explain why such genotypes do not correspond to the same phenotype when compared with individuals with the same genotype. As periodontitis is a multifactorial disease, both polymorphisms and DNA methylation may be associated. Example: an individual has a polymorphism associated with decreased gene expression in a certain gene and concomitantly has DNA hypomethylation (which is associated with increased gene expression) in that same gene. See that these two mechanisms have antagonistic effects and can explain why the analysis of a genotype alone does not fully explain the phenotype. We did not look for DNA methylation and have a different explanation consistent with our data (highlighted section 2.2, lines 195 – 219). We have also responded indirectly to this comment in the discussion in the highlighted last paragraph of section 3.1. (Lines 275 – 285).
- Authors: Section moved and Chinese reference clarified in the revised article, lines 78 - 81. Reviewer: The reference has been included
- Authors: Reference [2]is a study of IL1B-511 AA and GG alleles, but it is not clear how results of that study relate to our study. Unlike reference 2, we are not looking at IL1B-511 alleles by themselves, but rather in combination with alleles of IL1B+3954. Moreover, they looked only at homozygous C or T alleles (today called G or A alleles due to a new rule about direction of translation). We found that these homozygous common or variant alleles are uncommon in our EG study; 10 of our 15 hosts were heterozygous (A and G). Currently, we are preparing a proposal to determine whether we can compare the frequency of the weak GCF trait in Alzheimer’s patients with a similar group of non-Alzheimer’s control patients. Reviewer: Since the authors included an item on: How periodontitis relates to its comorbidities and emphasize dementia, a discussion at the level of polymorphisms associated with the two diseases is expected. Genetic studies based on the co-existence of periodontitis and systemic diseases are still scarce and need to be further explored. The proposal that the authors are preparing is interesting and could come with a perspective of the present work. In section 4.4, We added section 3.4. which is a discussion comparing IL1B-511(GG and GA) with IL1B-511(AA) effects on brain structures and their connectivity in Alzheimer’s disease patients. See lines 403 – 419.
Additional information: Many other changes were made to explain the results and their significance better, often as an indirect response from the reviewers. Perhaps the most important change was a recognition that the SNP alleles of IL10 at -597 were absent from reference 38, and that a PubMed search revealed no association with periodontal disease. The suggestion that IL10-597 was an alternative specifying gene for patient 10 was therefore removed from Tables 2, 3 and 5.
Two other important changes were to help the reader understand the abstract by revising the text to add rs numbers and also adding a concluding section 3.5, lines 420 – 431 to the Discussion. There, we note that our GCF phenotypic traits could be used to develop new methods for preventing or controlling periodontitis development. The revisions to the abstract and final section of the discussion are not highlighted.
We thank the reviewers for their constructive comments.
